# Face-to-Face Exercises Performed by Instructors to Improve the Mental Health of Japanese in the Community—A Randomized Control Trial

**DOI:** 10.3390/medicina56080404

**Published:** 2020-08-12

**Authors:** Akihiko Katayama, Kanae Kanda, Ayako Hase, Nobuyuki Miyatake

**Affiliations:** 1Faculty of Sociology, Shikoku Gakuin University, Zentsuji-shi, Kagawa 765-8505, Japan; 2Department of Public Health, Faculty of Medicine, Kagawa University, Kita-gun, Kagawa 761-0701, Japan; oda@med.kagawa-u.ac.jp; 3Department of Clinical Psychology, Faculty of Medicine, Kagawa University, Kita-gun, Kagawa 761-0701, Japan; ahase@med.kagawa-u.ac.jp; 4Department of Hygiene, Faculty of Medicine, Kagawa University, Kita-gun, Kagawa 761-0701, Japan; miyarin@med.kagawa-u.ac.jp

**Keywords:** face-to-face exercises, exercise using machines, mental health, self-efficacy, randomized control trial

## Abstract

*Background and objectives:* The purpose of this study is to clarify the effects on the mental health of face-to-face exercise performed by an instructor (lesson-style Group: Group L) and exercise using machines (program-style Group: Group P) by randomized control trial. *Materials and Methods:* Among 120 subjects, 117 subjects were allocated to two groups with stratified randomization by sex (Group P: 58 subjects; Group L: 59 subjects). A 60-min health exercise class was held once per week for 12 consecutive weeks. The measurement items were mental health as a primary evaluation item and self-efficacy as a secondary evaluation item. Physical fitness was also measured using a new physical fitness test used in Japan. The 12-item general health questionnaire (GHQ-12) was used to measure mental health and the general self-efficacy scale (GSES) was used to measure self-efficacy. *Results:* After the intervention, 102 subjects were analyzed. The changes in mental health evaluated by GHQ-12 scores were significantly lower in Group L −0.7 (95% CI, −1.2 to −0.3) than Group P −0.1 (95% CI, −0.4 to 0.2) (*p* = 0.03). The changes in self-efficacy evaluated by GSES scores were significantly higher in Group P 5.3 (95% CI, 3.1 to 7.5) than Group L 1.3 (95% CI, −0.4 to 3.1) (*p* < 0.01). *Conclusions:* Compared with program exercises mainly using machines, face-to-face exercises performed by instructors improved mental health.

## 1. Introduction

Physical activity as a health problem is being researched around the world. In 2010, the World Health Organization identified physical inactivity as the fourth leading risk factor for global mortality (6% of global deaths). Physical inactivity follows high blood pressure (13%), tobacco use (9%) and high blood glucose (6%). As a measure against physical inactivity, the “Global recommendations on physical activity for health” were announced [1].

Nine guidelines and four action areas were adopted as the “Toronto Charter for Physical Activity 2010” [2]. These guidelines require the use of science-based strategies to address the environmental and social determinants of physical activity and the need for a lifelong approach. In a special issue [3], The Lancet reported that 9.4% of all deaths in the world were due to lack of physical activity, the effect was comparable to obesity and smoking and “pandemic” worldwide [4]. In a second special issue, The Lancet reported that there was no significant change in the lack of physical activity in the world four years later [5].

In Japan, due to concerns about lack of exercise, various measures are being implemented to resolve the lack of exercise and to increase the amount of physical activity [6]. In particular, it is necessary to take measures against metabolic syndrome (MS) [7], which is a general metabolic disorder caused by abdominal obesity. Metabolic syndrome is a condition in which visceral fat-type obesity causes dyslipidemia, hyperglycemia and hypertension. It is often caused by a lack of exercise, overeating, etc., and improving lifestyle habits will prevent serious illness in the future. It is essential to improve your lifestyle.

Therefore, in Japan, a new health guidance system was started in 2008. This new system is called Specific Health Guidance. Specified health guidance is health support based mainly on the results of specific health checkups to prevent or eliminate metabolic syndrome. It is based on health guidance, emphasizing not only nutrition guidance, but also physical activity and exercise practices [8].

WHO lists physical, mental and social well-being as the three elements of health [9]. However, the Ministry of Health, Labor and Welfare reported that the total number of patients with mood disorders, such as depression, was 1,276,000, and the total number of patients has increased 2.9 times in the last 21 years [10]. The Ministry of Health, Labor and Welfare expects physical activity and exercise to improve mental health [6].

The effects of an exercise intervention on mental health were examined. Chen et al. examined the impact of a 6-month yoga program on the elderly by randomized control trial (RCT). After the intervention, older adults who participated in the yoga program had improved sleep quality and significantly improved depression compared to older adults who did not [11]. Antunes et al. conducted a cardiovascular intervention study in the elderly with RCT. Older people who participated in an ergometer aerobic fitness program every other day for six months had significantly lower depression and anxiety scores and improved quality of life than those who did not, although there was no significant change in the control group [12]. Blake et al. also reviewed 11 RCT-designed exercise interventions and reported that short-term positive results for depression or depressive symptoms were found in nine studies [13].

On the other hand, the effect on mental health due to the difference in the instructor’s teaching method during exercise has not been investigated.

In Japan, there are generally two methods used by the local government to conduct health exercise classes for community health. The first method is to install exercise equipment, such as an exercise bike, a walking machine and a simple strength training machine, in a local public facility and encourage participation in exercise programs. The second method is direct exercise guidance by an exercise leader and is carried out in a classroom format in a local public facility. To date, there are very few decision-making materials for the local government to use. Hence, to effectively use limited financial resources, it is imperative to select and provide appropriate physical activity/exercise to improve the mental health of residents.

The purpose of this study is to compare the effects of exercise instructors on the mental health of residents who participate in a face-to-face exercise classroom with those in mainly machine use classroom by randomized control trial (RCT).

## 2. Materials and Methods

### 2.1. Subjects

The recruitment of participants was carried out with the cooperation of the government (Zentsuji City). We asked the public relations magazine of the city to publish the contents of the offer. We asked the participants to mail us a request form. Then, we held a research participation briefing session and thoroughly explained the research contents. One hundred twenty people wanted to participate in the briefing session. After the briefing session, three people refused to participate. Two people were not attending because of their opposition from their family to join (personal circumstances). The other one did not reveal the reason for not participating (unknown).

A total of 117 community-dwelling Japanese subjects among 120 subjects, aged 67.2 (95% CI, 65.8 to 68.8) years, were enrolled in this RCT study. The subjects all met the following criteria: (1) they voluntarily took part in the study in May 2016 at Zentsuji city, Kagawa prefecture, Japan, (2) they had no limitation of exercise and (3) written informed consent was obtained.

Ethical approval was obtained from the ethical committee of Shikoku Gakuin University, Zentsuji city, Kagawa prefecture, Japan (approval number: 2015001, approval date: 15 April 2015).

### 2.2. Study Design

The design of this study is a Randomized Controlled Trial (RCT) (Figure 1). The 117 enrolled subjects were allocated to two groups: a program-style group that mainly used machine exercise (Group P: 58 subjects) and a lesson-style group that took part in face-to-face exercise (Group L: 57 subjects) with stratified randomization by sex.

Group P consisted of groups in which participants exercised voluntarily using exercise equipment. The instructor in charge carried out instruction on the use of equipment, exercise program creation and safety management. The health exercise class consisted of a mixture of conditioning, aerobic exercise and strength training. Instructors personally responded to questions regarding the use of equipment and items regarding the implementation of the program. A 60-min health exercise class was held once per week for 12 consecutive weeks.

Group L carried out health exercises with face-to-face, direct guidance from instructors. The instructor in charge held a 60-min lesson-style health exercise class. The content of the class was a combination of conditioning, aerobic exercise and strength training. The 60-min healthy exercise class was held once per week for 12 consecutive weeks.

The role of the instructor varies significantly between groups. In Group P, the instructor’s main role when instructing exercises was safety management and answering questions regarding exercise programs. We distributed the exercise program individually and set the situation so that participants could complete the exercise program within the set time. In Group L, the main role of the instructor in the exercise instruction was direct in-person exercise instruction. We set the environment for participants to exercise during the set time by face-to-face instruction from the instructor. Administrative expenses and operating costs were equal for both groups.

Measurements were performed before and after the intervention. We performed measurements before and after the intervention. All baseline measurements were completed one week before the intervention. All measurements were completed one week after the end of the intervention. The measurement items were mental health as a primary evaluation item and self-efficacy as a secondary evaluation item. Physical fitness change was also measured.

### 2.3. Sample Size

The primary endpoint was mental health. We used GHQ-12 as the measurement scale. We compared the change in measured values before and after the intervention between the two groups. We calculated the sample size using the significance level, power, difference between the two groups and standard deviation (SD) in an unpaired t-test. Following similar previous studies, the significance level used was α: 0.05; the power used was 0.8.

The difference between the two groups was derived as follows: S Taneichi et al. examined the effect of walking 10,000 steps per day on depressive symptoms for Japanese [14]. They reported that the Group that achieved 600,000 steps in 60 days (*n* = 125) had a significantly reduced GHQ-12 score compared to the Group that did not (*n* = 46). In addition, they reported that the GHQ-12 score of the Group that achieved the specified number of steps changed from baseline (2.16 ± 2.41) to post-intervention (1.42 ± 1.78). After 60 days of intervention, the change in the GHQ-12 score was about 0.7. Based on this study, we set the change in the GHQ-12 score after the intervention as 0.7.

We used a GHQ-12 score that was regularly measured in a group of participants (*n* = 38, 26.3% men) in a health exercise class held in the community. The GHQ-12 score change over a year was −0.1 ± 1.2. Therefore, the SD of change in GHQ-12 score was set to 1.2 in this study.

The sample size test consisted of an unpaired t-test (significance level: α = 0.05, power: power = 0.8, difference to be detected: 0.7, standard deviation: 1.2, ratio of the number of cases: 1). The sample size was calculated using the JMP ^®^ 12 software (SAS Institute, Japan, Tokyo, Japan). The required number of samples was 95. We anticipated a dropout rate of 10% and set the recruitment size at 105.

### 2.4. Clinical Parameters and Measurements.

The following clinical parameters were evaluated: age, sex, height, body weight and body-fat percentage. Body-fat percentage was measured with a body composition meter (MC-180: TANITA, Tokyo, Japan). Each patient’s body mass index (BMI) was calculated as follows: body weight (kg)/[height (m)]^2^.

### 2.5. Mental Health Assessment

GHQ-12 is a useful and reliable tool for measuring psychological distress in the general adult population [15,16]. In this study, we used the Japanese version of GHQ-12. This survey consisted of 12 questions that examined the patient’s general well-being, concentration, decision-making, strain, problem-solving, self-confidence and self-worth in the past few weeks [16]. The answer was organized into four-stage, as follows: “much less than usual”, “same as usual”, “more than usual”, and “much more than usual”. In this study, the GHQ method was used as the scoring method. This method is the standard scoring method Goldberg recommends identifying cases. In this method, the two least symptomatic answers are scored as 0 and the two most symptomatic answers are scored as 1 (0–0-1–1). GHQ-12 has a minimum total score of 0 and a maximum GHQ-12 total score of 12 [16]. Higher scores were considered lower mental health, and cutoff scores of three or less were considered to reflect good mental health [17].

### 2.6. Self-Efficacy

A concept called self-efficacy was proposed by Bandura [18]. In social learning theory, three factors, a precedent factor, a result factor and a cognitive factor, are considered as the factors that decide the actions of a person. Self-efficacy is the central element of the precedent factor [18]. GSES (general self-efficacy scale) measures personal general self-efficacy with high reliability and validity, as previously described [19,20]. Subjects with a higher GSES score work harder, are more tolerant and can adapt their behavior [21].

The GSES was created by Sakano to measure the strength of general self-efficacy, which is how much individuals generally tend to perceive self-efficacy as high or low [22]. GSES consists of 16 questions to measure the degree of general self-efficacy. The answer is a two-point method of “yes” or “no” and the score range is 0 to 16 points. The score obtained here is used as a standardized score using a standardized score conversion table. The standardized score is the same as the deviation value, with an average of 50 and a standard deviation of 10. The higher this standardization score, the higher the self-efficacy can be considered [22].

### 2.7. Physical Fitness

Physical fitness was evaluated by grip strength (kg), flexibility (cm), standing on one leg with eyes open (seconds) and a six-minute walk (meters). All items were measured based on the new fitness test of the Ministry of Education, Culture, Sports, Science and Technology-Japan [23,24].

Grip strength measurements evaluated muscle strength. A Smedley grip dynamometer was used. Two measurements were taken for each of the left and right sides. The record with the higher numeric value for both sides was averaged and adopted [23,24].

Body flexion measurements evaluated flexibility. Sit with your back on the wall, with your legs extended and without bending your knees, bend your body forward. A measurement object with a height of 24 cm was pushed out with both thumbs and the moving distance of the measurement object was measured. Two measurements were taken and the better one was adopted [23,24].

The balance ability was measured by standing on one foot with the eyes open. While standing, place both hands on your hips and raise one leg and measure the duration. Two measurements were taken and the better one was adopted [23,24].

The cardiorespiratory endurance was measured by walking for 6 min. Walk for six minutes at a normal walking speed, with either foot always on the ground. The distance walked in 6 min was measured. The measurement was performed only once [23,24].

### 2.8. Statistical Analysis

The data were expressed as the mean, 95% confidence intervals (95% CI) of consecutive variables and the number of variables classified (%). An unpaired t-test was used to compare the two groups, where *p* < 0.05 was significant. Statistical analysis was performed using JMP^®^14 software (SAS Institute, Japan, Tokyo, Japan).

## 3. Results

### 3.1. Clinical Characteristics of Enrolled Participants at Baseline

Table 1 summarizes the baseline clinical profile of 117 people who wanted to participate in the study after the research participation briefing session.

For the 12 weeks of exercise, five participants in Group P and five participants in Group L had a low participation rate (<80%), and three participants in Group P and two participants in Group L dropped out. Therefore, for the final analysis, we evaluated 50 participants in Group P and 52 participants in Group L. The clinical profiles of the enrolled participants at baseline are summarized in Table 2.

### 3.2. Comparison of the Changes in Parameters between the Two Groups

We compared changes in the parameters between the two groups (Table 3). In the change in the physical fitness index, there was a significant difference only in flexibility. There were no significant differences in the changes in grip strength, standing on one leg with eyes open and distance walked indices.

The changes in mental health evaluated by GHQ-12 scores in Group L -0.7 (95% CI, −1.2 to −0.3) were significantly lower than those in Group P −0.1 (95% CI, −0.4 to 0.2) (*p* = 0.03). In addition, the changes in self-efficacy evaluated by GSES scores in Group P 5.3 (95% CI, 3.1 to 7.5) were significantly higher than those in Group L 1.3 (95% CI, −0.4 to 3.1) (*p* < 0.01).

## 4. Discussion

In this study, we compared lesson-style exercises with program-style exercises. Mental health improved significantly in the lesson group (Group L) compared to the program group (Group P). Self-efficacy was significantly improved in Group P compared to Group L. In terms of physical fitness, only flexibility was significantly improved in Group P compared to Group L.

There are many reports on the physical effects of exercise. For example, the American College of Sports Medicine provides exercise guidelines that are effective for physical health and fitness.

As for the effect of exercise on mental health, as shown in the introduction, its effect was verified. However, the methodology of providing exercise has not been verified.

Even in Japan, measures for mental health are being fully considered. In national policy, mental health is also emphasized by the Health Japan 21 concept. The goal of this political measures is to reduce the proportion of those who have psychological distress equivalent to mood disorder/anxiety disorder to 9.4% in 2022 (2000 at the time of formulation: 10.4%).

In this study, face-to-face exercises performed by an instructor improved mental health scores compared to exercises involving machine exercises. It is possible that the face-to-face exercise contributed to the improvement of mental health through the interaction between participants and invocation from the instructor during the lesson. On the other hand, active participation in self-exercise, such as achievement of a given program and routines, contributed to the improvement of self-efficacy. Therefore, it is crucial to select the method of providing exercise according to its purpose.

It is essential to consider not only mental health, but self-efficacy. McAuley et al. conducted a 24-month prospective cohort study in community-dwelling older adults and showed a relationship between physical activity, self-efficacy and QOL (quality of life) [25]. Through physical and mental health, self-efficacy is improved, and, consequently, QOL.

In Japan, many local governments run health classes to maintain and improve the health of residents that provide physical education facilities managed by the local government, face-to-face exercises by instructors in the studio and exercise mainly by machine in the training room. Therefore, the administration should clarify its purpose and consider how the exercise should be provided.

There are some limitations to our research. First, since the duration of the intervention was only 12 weeks, we could not evaluate the long-term effects of face-to-face exercises performed by instructors. Second, we evaluated only 52 participants in Group L and 50 participants in Group P, therefore, the statistical power may be slightly lower than the initial estimation. Third, the exercise load was not universal among the intervention groups. Fourth, Enrolled subjects in this study were thought to be more health-conscious than the average person.

## 5. Conclusions

Face-to-face exercises performed by instructors improve mental health compared with program exercises mainly using machines. Conversely, program exercises primarily using machines improve self-efficacy compared with face-to-face exercises by instructors.

## Figures and Tables

**Figure 1 medicina-56-00404-f001:**
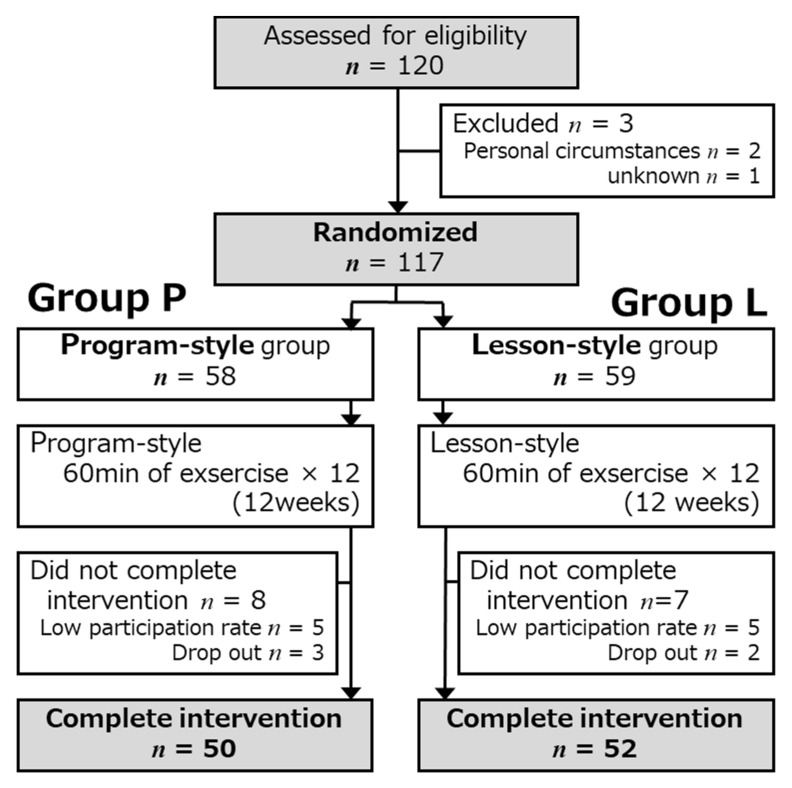
Flow chart of the study design.

**Table 1 medicina-56-00404-t001:** Characteristics of subjects before randomized.

	All Subjects (*n* = 117)			
	Mean	95%CI		
Male, n (%)	17 (14.5%)			
Age (year)	67.3	65.7	-	68.8
Height (cm)	155.3	154.0	-	156.6
Body weight (kg)	54.1	52.7	-	55.5
BMI (kg/m^2^)	22.4	21.9	-	22.9
Body fat (%)	27.1	25.6	-	28.6
Grip strength (kg)	26.7	25.4	-	27.9
Flexibility (cm)	39.6	38.1	-	41.0
One leg with eye open balance (seconds)	82.2	74.8	-	89.6
Six-minute walking (m)	524.0	512.5	-	535.5
GHQ-12 score	1.0	0.7	-	1.3
GSES score	50.9	49.1	-	52.6

BMI: Body mass index (kg/m^2^); GHQ-12: General Health Questionnaire-12; GSES: General Self-Efficacy Scale; CI: confidence intervals.

**Table 2 medicina-56-00404-t002:** Baseline characteristics of subjects.

	P: Program-Style Group (Machines) (*n* = 50)				L: Lesson-Style Group (Instructor) (*n* = 52)			
	Mean	95%CI			Mean	95%CI		
Male, n (%)	5 (10.0%)				6 (11.5%)			
Age (year)	66.8	64.6	-	69.0	67.9	65.9	-	70.0
Height (cm)	153.8	152.2	-	155.5	155.8	153.8	-	157.8
Body weight (kg)	52.6	50.7	-	54.5	54.7	52.6	-	56.7
BMI (kg/m^2^)	22.2	21.5	-	22.9	22.5	21.8	-	23.3
Body fat (%)	27.2	25.1	-	29.3	27.6	25.4	-	29.8
Grip strength (kg)	26.4	24.8	-	28.0	26.2	24.1	-	28.3
Flexibility (cm)	40.3	38.3	-	42.3	39.1	36.6	-	41.6
One leg with eye open balance (seconds)	84.6	72.8	-	96.5	84.4	73.1	-	95.6
Six-minute walking (m)	534.2	515.0	-	553.4	516.7	502.4	-	531.1
GHQ-12 score	0.9	0.6	-	1.3	1.0	0.6	-	1.5
GSES score	50.4	47.7	-	53.2	51.0	48.2	-	53.7

BMI: Body mass index (kg/m^2^); GHQ-12: General Health Questionnaire-12; GSES: General Self-Efficacy Scale.

**Table 3 medicina-56-00404-t003:** Comparison of changes between groups.

	P: Program-Style Group (Machines) (*n* = 50)				L: Lesson-Style Group (Instructor) (*n* = 52)				
	Mean	95%CI			Mean	95%CI			*p* Value
Grip strength (kg)	−1.1	−1.7	-	−0.4	−0.4	−1.4	-	0.6	0.29
Flexibility (cm)	2.2	0.5	-	3.8	−0.5	−2.3	-	1.4	**0.03**
One leg with eye open balance (seconds)	7.5	−1.9	-	16.9	2.9	−3.8	-	9.7	0.42
Six-minute walking (m)	48.7	32.7	-	64.7	50.6	38.3	-	62.9	0.85
GHQ-12 score	−0.1	−0.4	-	0.2	-0.7	−1.2	-	−0.3	**0.03**
GSES score	5.3	3.1	-	7.5	1.3	−0.4	-	3.1	***p* < 0.01**

GHQ-12: General Health Questionnaire-12; GSES: General Self-Efficacy Scale; Bold values are statistically significant (*p* < 0.05).

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
