# Peer review of "Face-to-Face Exercises Performed by Instructors to Improve the Mental Health of Japanese in the Community—A Randomized Control Trial"

_medicina, 2020, doi:10.3390/medicina56080404_

Round 1

Reviewer 1 Report

Confidence intervals would be a better statistic to report in the abstract rather than +/-

Final sentence in the abstract is not evidenced by what else has been written in the abstract. Cost has not been mentioned in the abstract, neither has purpose or appropriateness of exercise and the use of 'effective' is not clear - effective in terms of mental health or physical health? This last sentence should either be removed or evidence for it should be provided in the rest of the abstract.

I don't understand what the percentages are in the first para of the introduction. 

The introduction is not very clear - for example, what exactly is metabolic syndrome and are the 'variety of measures' being implemented in Japan?

Introduction is very poor - there is a vast amount of research examining the impact of exercise on mental health, none of this is reported. The authors have not given any background on why it's important to compare lesson based exercise with machine exercise - why do the authors think it would be different, why is this important, what other work has been done in this area?

Methods:

Where was the sampled drawn from, why these 117? How were they recruited, why did 3 drop out/not consent?

If group P had an instructor present at all times, why and how is this different from group L?

The way in which the minimal clinically important difference is unusual - it's been based on a another trial with only 171 participants - this is a very small sample. 

Description of GHQ-12 is not clear - what is the 'scoring method'? Please provide full psychometric properties for the measure, was it translated from English, if so, has this translation been validated? Same comments for the GSES. More information needed about the physical fitness tests too.

How often were the outcomes measured? Why was no pre-test measurement made? And if it was, why did you use an unpaired t-test?

Table 1: You should not only include the baseline characteristics of participants that didn't drop out - it's vital that ALL participants are included in this table. I don't think having min and max scores is necessary.

Baseline GHQ-12 scores are very low in both groups - why was this population chosen? 

The difference in GHQ-12 is minimal and I would query its clinical value. What are the 'changes between groups' you are referring to in table 2, reporting confidence intervals is a better way to present your data.

The 'changes in mental health' are NOT significantly lower in Group L, the change in mental health is greater in Group L. Your results also show that self-efficacy was improved more in Group P. So what you're showing is that Group L had more improved mental health, but Group P had more improved self-efficacy. 

Discussion:

Lines 166-172 should have a reference to evidence what is being said.

Why are so many papers being introduced in the discussion, these should have been presented in the introduction.

Once again, the assumption that face-face exercises improve mental health more than self-exercise can not be supported since the self-exercise also had face-to-face instructors present. The exercises being compared where completely different physically with the classroom ones relying more on aerobic activity. This has briefly been mentioned in the limitations but I think this is a major limitation and should be discussed in much more detail.

Given my comments above I feel that the conclusions drawn are unfounded.

Author Response

Response to Reviewer 1 Comments

Thank you for your review of our article. We have answered each of your points below.

Major comments

Abstract

Introduce

Point 1: Confidence intervals would be a better statistic to report in the abstract rather than +/-

Response 1:

I understood that the expression by the confidence interval is more appropriate than the expression by the standard deviation. We have modified the standard deviation expression to a confidence interval expression. Please check the revised parts of the abstract.

Point 2: Final sentence in the abstract is not evidenced by what else has been written in the abstract. Cost has not been mentioned in the abstract, neither has purpose or appropriateness of exercise and the use of 'effective' is not clear - effective in terms of mental health or physical health? This last sentence should either be removed or evidence for it should be provided in the rest of the abstract.

Response 2:

In the abstract, we understood that the last sentence was irrelevant. We have deleted the last sentence.

Point 3: I don't understand what the percentages are in the first para of the introduction.

Response 3:

As you pointed out, we were only displaying the percentage numbers. This expression was very confusing. We have included the meaning of the percentage figures in the text. In addition, we have modified the text to be as easy to understand as possible.

Point 4: The introduction is not very clear - for example, what exactly is metabolic syndrome and are the 'variety of measures' being implemented in Japan?

Response 4:

We understand the matter you pointed out.

The explanation of the metabolic syndrome was inadequate, and the description of the measures of the Japanese government was also lacking. Considering these things, we have corrected the text with some explanation. Please check the revised manuscript.

Using the example of metabolic syndrome, we have explained the need for government health measures to address the lack of physical activity and reduce activity.

Point 5: Introduction is very poor - there is a vast amount of research examining the impact of exercise on mental health, none of this is reported. The authors have not given any background on why it's important to compare lesson based exercise with machine exercise - why do the authors think it would be different, why is this important, what other work has been done in this area?

Response 5:

Thank you for your advice. Added a paper that reports that exercise affects mental health. Please check the revised manuscript.

The effects of exercise on mental health have been reported. In addition, the content of exercises such as aerobic exercise and resistance exercise has also been reported. However, there are no research reports on methods of exercise guidance. For us, movement leaders, how to convey the movements to the participants, is a very important factor. There is a big difference between face-to-face exercise instruction and exercise instruction only with a program that shows the exercise content. From the experience of many exercise instruction, we experienced the effect of mental health by the difference in the method of providing exercise. However, there is no basis for this, and it has not been reported from past research. We found it essential to clarify the effectiveness of the face-to-face exercise classroom and the effectiveness of the programmed exercise classroom. We think that this consideration is important, especially in the current Corona (COVID-19) related confusion, although there are many exercise instructions options.

Methods:

Point 6: Where was the sampled drawn from, why these 117? How were they recruited, why did 3 drop out/not consent?

Response 6:

We understand the matter you pointed out. Regarding the recruitment of participants, we did not provide a clear explanation in the article.

The recruitment of participants was carried out with the cooperation of the government (Zentoji City). We asked the public relations magazine of the city to publish the contents of the offer. We asked the participants to mail us a request form. Then, we held a research participation briefing session and thoroughly explained the research contents. One hundred twenty people wanted to participate in the briefing session. After the briefing session, three people refused to participate. Two people were not attending because of their opposition from their family to join (personal circumstances). The other one did not reveal the reason for not participating (unknown).

The above contents are summarized and described in the text of the paper. Please check the article.

Point 7: If group P had an instructor present at all times, why and how is this different from group L?

Response 7:

Thank you for your question.

There was insufficient explanation in the paper of the differences between the groups. As you pointed out, instructors are resident in both groups. However, their roles are very different.

In Group P, the instructor's main role when instructing exercises was safety management and answering questions regarding exercise programs. We distributed the exercise program individually and set the situation so that participants could complete the exercise program within the set time.

In Group L, the main role of the instructor in the exercise instruction was direct in-person exercise instruction. We set the environment for participants to exercise during the set time by face-to-face instruction from the instructor.

The above contents are also added to the article. Please check the text of the article.

Point 8: The way in which the minimal clinically important difference is unusual - it's been based on a another trial with only 171 participants - this is a very small sample.

Response 8:

In this study, we calculated the sample size. As a result of the calculation, 95 subjects were required in this study. From our experience so far, we have set the dropout rate to 10%. The required number of subjects in this study was 105. The number of subjects analyzed after the intervention was 102. Since the number of samples at the planning stage was 95, the number of subjects to analysis was 107.3%.

Point 9: Description of GHQ-12 is not clear - what is the 'scoring method'? Please provide full psychometric properties for the measure, was it translated from English, if so, has this translation been validated? Same comments for the GSES. More information needed about the physical fitness tests too.

Response 9:

Thank you for your advice.

We had added a description of the measurement content to each measurement item. Please check the text of the article.

Point 10: How often were the outcomes measured? Why was no pre-test measurement made? And if it was, why did you use an unpaired t-test?

Response 10:

Thank you for pointing out.

We found that it was an unclear statement about the measurement situation and the timing of measurement. We understood that the explanation was inadequate.

We performed measurements before and after the intervention. All baseline measurements were completed one week before the intervention. All measurements were completed one week after the end of the intervention.

In terms of statistical processing, baseline and endpoint changes were compared between groups. Therefore, we have adopted the unpaired T-test.

We have added to the paper a statement about the timing of the measurement implementation. Please check the text in the article.

Regarding the pre-test, we were not able to pre-test the subjects. The period of use of research funds was limited, and it took only about one month from the subjects decides to the start of the intervention.

Point 11: Table 1: You should not only include the baseline characteristics of participants that didn't drop out - it's vital that ALL participants are included in this table. I don't think having min and max scores is necessary.

Response 11:

Thank you for your advice.

We have created a new table. Therefore, we have prepared three tables. Table 1 shows data for 117 randomized people. We displayed baseline data before a few dropped out. Table 2 shows baseline data by group. Table 3 is a comparison between groups of measured changes after intervention.

Point 12: Baseline GHQ-12 scores are very low in both groups - why was this population chosen?

Response 12:

Regarding the recruitment of participants, we have posted the recruitment information in the local government newsletter with the cooperation of the local government. We were unable to prepare transportation for the participants. Participants were limited to those who could secure public transportation, private cars, bicycles, and other means of transportation to the venue. Therefore, it is considered that relatively active people participated.

Point 13: The difference in GHQ-12 is minimal and I would query its clinical value. What are the 'changes between groups' you are referring to in table 2, reporting confidence intervals is a better way to present your data.

Response 13:

Thank you for your advice.

GHQ-12 scores improved by an average of 0.7 before and after the 12-week intervention. The fact that the GHQ-12 score improved by 0.7 means that a little less than one of the question items in GHQ-12 improved. From this point, it may be considered of low clinical value. The short 12-week period may have influenced the outcome of this intervention.

Since the subjects were interested in exercise classes and recruited themselves this time, so It is possible that they were more health-conscious than the general public. We think that it was also due to good mental health at the baseline.

We have included this in the Limitation part of our research.

Point 14: The 'changes in mental health' are NOT significantly lower in Group L, the change in mental health is greater in Group L. Your results also show that self-efficacy was improved more in Group P. So what you're showing is that Group L had more improved mental health, but Group P had more improved self-efficacy.

Response 14:

This study was designed with the view that mental health and self-efficacy are two completely different dimensions. As you pointed out, we felt that self-efficacy could be considered as a part of mental health or a comprehensive matter. So, we checked a research paper on mental health and self-efficacy

Schönfeld et al. reported that general self-efficacy is a mediator of stress affecting positive and negative mental health [1]. So self-efficacy may be affecting mental health.

We found that future research would require careful consideration of the relationship between mental health and self-efficacy.

In this research, we would like to proceed with the article on the assumption that mental health and self-efficacy are different matters.

  1. Schönfeld, P.; Brailovskaia, J.; Bieda, A.; Zhang, X.C.; Margraf, J. The effects of daily stress on positive and negative mental health: Mediation through self-efficacy. International Journal of Clinical and Health Psychology 2016, 16, 1–10, doi:10.1016/j.ijchp.2015.08.005.

Discussion:

Point 15: Lines 166-172 should have a reference to evidence what is being said.

Response 15:

Thank you for your advice.

We have described the physical and mental effects to emphasize the mental effects of exercise. However, as you pointed out, it did not show a clear basis, and it was an unclear expression. We found that this part was hard to understand. We have deleted the section that describes the physical effects.

Point 16: Why are so many papers being introduced in the discussion, these should have been presented in the introduction.

Response 16:

We have shown in the introduction that the effects of exercise on the mental side have been demonstrated. But that the differences in the methods of exercise instruction are not yet clear. From there, we sought to guide the need for this study.

In the discussion, the purpose of posting the literature on the effect of exercise on the mental side again is as follows. Many studies have been conducted to examine the psychological effects of exercise intervention, but the methodology of exercise instruction has not been discussed. To emphasize that, we have presented a more specific intervention study.

Point 17: Once again, the assumption that face-face exercises improve mental health more than self-exercise can not be supported since the self-exercise also had face-to-face instructors present. The exercises being compared where completely different physically with the classroom ones relying more on aerobic activity. This has briefly been mentioned in the limitations but I think this is a major limitation and should be discussed in much more detail.

Response 17:

Instructors are resident in Group L and Group P, but their roles are very different. We explain it in point 7 of the question.

In Group P, the instructor's main role when instructing exercises was safety management and answering questions regarding exercise programs. We distributed the exercise program individually and set the situation so that participants could complete the exercise program within the set time.

In Group L, the main role of the instructor in the exercise instruction was direct in-person exercise instruction. We set the environment for participants to exercise during the set time by face-to-face instruction from the instructor.

In this research, we would like to discuss not the instructor's permanent residence, but the form of instructor's exercise instruction.

Regarding exercise load, it was pointed out that the exercise load was not uniform among the groups. We have not been able to equalize the exercise load completely, but we have taken steps to make it as uniform as possible.

In the present study, two principal instructors were alternately responsible for the interventions of the two groups. In the Lesson-style group (Group L), direct exercise instruction was given. In the Program-style group (Group P), the instructor created a training program using machines and conducted safety management. There are two reasons why the instructors took turns intervening in the two groups. Firstly, to eliminate the bias caused by the difference in instructors, the instructors intervened while taking turns. Secondly, to equalize the exercise load, the instructors intervened in alternation.

Generally, exercise prescription is composed of exercise frequency, intensity, time, and type (ACSM's Guidelines for Exercise Testing and Prescription, eighth edition, American College of Sports Medicine, 2009). In this intervention, the frequency and time of exercise were uniforms. Type is also the most critical aspect of intervention in this study. The exercise intensity was made uniform by the teaching technique and the exercise program making technique by the instructors. The instructors were qualified for exercise guidance issued by the Japan Health Promotion Fitness Foundation, an affiliated organization operating under the policies of the Japanese Ministry of Health, Labor and Welfare. Therefore, the instructors who intervened this time had sufficient teaching skills.

The limitation of research indicates that the exercise load could not be equalized entirely.

Point 18: Given my comments above I feel that the conclusions drawn are unfounded.

Response 18:

Thank you for your advice.

Please consider the role of the instructor in each group again. (Reference: points 7 and 17)

Due to our lack of expression, the content of this research was not accurately transmitted.

We sincerely apologize for the inconvenience to the referee's teachers by having peer-reviewed such a paper with insufficient information.

The lack of information regarding the role of the instructor utterly lacked in our explanation. It turned out that the description of the role of the instructor is an essential point of this paper.

As you pointed out, we were able to add the missing information to the paper. I think there are still deficiencies, but we would appreciate your kind guidance.

Reviewer 2 Report

The purpose of this study is to clarify the effects on the mental health of face-to-face exercise performed by an instructor (lesson-style group: Group L) and exercise using machines (program-style group: Group P). The authors present a randomized control trial with the following objetive:

to compare the effects of exercise instructors on the mental health of residents who participate in a face-to-face exercise classroom with those in mainly machine use classroom by randomized control trial (RCT).

In regards to this objective, the authors found that the changes in mental health evaluated by GHQ-12 scores were significantly lower in Group L (-0.7 ± 1.6) than Group P (-0.10 ± 21 1.19) (p = 0.03).

Major comments

In an interventional clinical trial blinding and randomization are particularly important to assess the results. The authors show two groups with stratified randomization by sex. Participants in the study are 10% male and 90% female. Could the authors explain the reasons to stratify by sex? Could the authors explain the implementation of the randomization? Who generated the random allocation sequence, who enrolled participants, and who assigned participants to interventions.

The authors state that the intervention was a 60-minute health exercise class once a week for 12 consecutive weeks, one group performed by an instructor and the other exercise using machines. I consider it interesting to show the results of physical practice of both groups, at baseline and during the intervention, to rule out that it is acting as a confounding factor. Could clarify the limitation: “the exercise load was not universal among the intervention groups”?.

Minor comments

On line 45, the acronym MS is use only once, maybe is not necessary an acronym in this case.

Could the lacks of blinding in this intervention affect to the results? Could the motivation have been greater in the group in contact with the instructor?

Perhaps the tables would be more self-explanatory if specified Program-style group (machines) and Lesson-style group (instructor).

Author Response

Response to Reviewer 2 Comments

Thank you for your review of our article. We have answered each of your points below.

Major comments

Point 1: In an interventional clinical trial blinding and randomization are particularly important to assess the results. The authors show two groups with stratified randomization by sex. Participants in the study are 10% male and 90% female. Could the authors explain the reasons to stratify by sex? Could the authors explain the implementation of the randomization? Who generated the random allocation sequence, who enrolled participants, and who assigned participants to interventions.

Response 1:

We understand the matter you pointed out.

The subjects in this study were 10% male and 90% female, and there was a possibility that gender differences could occur between the groups when simple randomization was performed.

The measurement items were mental health as the primary evaluation item and self-efficacy as the secondary evaluation item.

In a survey of mental health and self-affirmation, we determined that sex was an influencing factor and adopted stratified randomization by sex.

Stratified randomization was performed by random number generation by PC.

Research supporter A generated a random number from a PC.

Research supporter B registered the participant using the random number.

Finally, Assistant C assigned participants to the intervention site.

Point 2: The authors state that the intervention was a 60-minute health exercise class once a week for 12 consecutive weeks, one group performed by an instructor and the other exercise using machines. I consider it interesting to show the results of physical practice of both groups, at baseline and during the intervention, to rule out that it is acting as a confounding factor. Could clarify the limitation: “the exercise load was not universal among the intervention groups”?

Response 2:

Thank you for your question.

In the present study, two principal instructors were alternately responsible for the interventions of the two groups. In the Lesson-style group (Group L), direct exercise instruction was given. In the Program-style group (Group P), the instructor created a training program using machines and conducted safety management. There are two reasons why the instructors took turns intervening in the two groups. Firstly, to eliminate the bias caused by the difference in instructors, the instructors intervened while taking turns. Secondly, to equalize the exercise load, the instructors intervened in alternation.

Generally, exercise prescription is composed of exercise frequency, intensity, time, and type (ACSM's Guidelines for Exercise Testing and Prescription, eighth edition, American College of Sports Medicine, 2009). In this intervention, the frequency and time of exercise were uniforms. Type is also the most critical aspect of intervention in this study. The exercise intensity was made uniform by the teaching technique and the exercise program making technique by the instructors. The instructors were qualified for exercise guidance issued by the Japan Health Promotion Fitness Foundation, an affiliated organization operating under the policies of the Japanese Ministry of Health, Labor and Welfare. Therefore, the instructors who intervened this time had sufficient teaching skills.

Minor comments

Point 3: On line 45, the acronym MS is use only once, maybe is not necessary an acronym in this case.

Response 3:

Thank you for your advice. We have removed the "MS" notation.

Point 4: Could the lacks of blinding in this intervention affect to the results? Could the motivation have been greater in the group in contact with the instructor?

Response 4:

Thank you for your question.

The intervention is clear to the participants and not blind. However, the specific differences between groups of interventions are not informed to participants. Therefore, participants cannot make comparisons between groups.

From the above, it is considered that there was no effect on the results due to the fact that the exercise method could not be blinded. Therefore, I don't think the lesson group was motivated.

Point 5: Perhaps the tables would be more self-explanatory if specified Program-style group (machines) and Lesson-style group (instructor).

Response 5:

Thank you for your guidance. In order to make it easier to understand in the notation of group name in the table,

Added (instructors) and (machine) as below.

L: Lesson-style group (instructors)

P: Program-style group (machine)

We indicated the corrections in the article in red.

Due to our lack of expression, the content of this research was not accurately transmitted.

We sincerely apologize for the inconvenience to the referee's teachers by having peer-reviewed such a paper with insufficient information.

As you pointed out, we were able to add the missing information to the paper. I think there are still deficiencies, but we would appreciate your kind guidance.

Round 2

Reviewer 2 Report

I agree with the present manuscript.